# Human Milk Microbiome from Polish Women Giving Birth via Vaginal Delivery—Pilot Study

**DOI:** 10.3390/biology14040332

**Published:** 2025-03-25

**Authors:** Agnieszka Chrustek, Agnieszka Dombrowska-Pali, Dorota Olszewska-Słonina, Natalia Wiktorczyk-Kapischke, Maciej W. Socha, Anna Budzyńska, Iwona Sadowska-Krawczenko

**Affiliations:** 1Department of Pathobiochemistry and Clinical Chemistry, Faculty of Pharmacy, L. Rydygier Collegium Medicum in Bydgoszcz, Nicolaus Copernicus University in Toruń, M. Curie-Skłodowska 9 St., 85-094 Bydgoszcz, Poland; a.chrustek@cm.umk.pl; 2Department of Perinatology, Gynecology and Gynecological Oncology, Faculty of Health Sciences, L. Rydygier Collegium Medicum in Bydgoszcz, Nicolaus Copernicus University in Toruń, Łukasiewicza 1 St., 85-821 Bydgoszcz, Poland; agnieszka.pali@cm.umk.pl (A.D.-P.); maciej.socha@cm.umk.pl (M.W.S.); 3Department of Microbiology, Faculty of Pharmacy, L. Rydygier Collegium Medicum in Bydgoszcz, Nicolaus Copernicus University in Toruń, M. Curie-Skłodowska 9 St., 85-094 Bydgoszcz, Poland; natalia12127@gmail.com (N.W.-K.); a.budzynska@cm.umk.pl (A.B.); 4Department of Neonatology, Faculty of Medicine, L. Rydygier Collegium Medicum in Bydgoszcz, Nicolaus Copernicus University in Toruń, Ujejskiego 75 St., 85-168 Bydgoszcz, Poland; iwonasadowska@cm.umk.pl

**Keywords:** microbiome, human milk, vaginal delivery, breastfeeding, culture-based method

## Abstract

Breast milk is one of the most important factors shaping the bacterial flora of the digestive system of a newborn and infant. To date, numerous studies have been conducted on the source and factors influencing the diversity of the composition of breast milk bacteria. Currently, there are three hypothetical sources of the breast milk microbiome. The first is based on an endogenous mechanism (intracorporeal translocation of bacteria from the mother’s intestinal microflora to the mammary gland, using dendritic cells). The second route is the transfer of mother’s bacteria from the mother’s skin located on the areola and nipple during breastfeeding. The third hypothesis indicates the reverse flow of breast milk from the child’s oral cavity to the woman’s milk ducts. In order to assess the breast milk microbiome, as well as the microbiological assessment of the skin I confirmof the areola and nipple before and after breastfeeding, 86 women with normal BMI, originating from Poland, and giving birth at term vaginally, were included in the study. Each participant in the study provided 40 mL of milk and a swab from the areola and nipple before and after breastfeeding. In order to culture and identify microorganisms from the collected material, appropriate culture methods and media were selected. A total of 120 species of bacteria were isolated, mainly from the genera *Streptococcus* and *Staphylococcus*, which constitute the basic composition of the microbiota of human milk. In addition, species specific only to human milk were identified.

## 1. Introduction

Human milk (HM) is defined as a biological fluid with a dynamic composition, in which not only nutrients but also bioactive components are distinguished, including, among others, cytokines, immunoglobulins, bioactive lipids, human milk oligosaccharides (HMOs), hormones, and microorganisms [1]. Due to the fact that human milk adapts to the needs and health status of the child, it is considered a unique food that meets all the needs of the child. The composition of human milk changes in response to factors such as date of delivery, body mass index (BMI) of the mother, duration of lactation, and time of day. Studies are increasingly indicating that the mother’s diet also has a significant impact on the composition of human milk. Ward et al. showed that increased consumption of sugar and fat changed the concentrations of triglycerides, cholesterol, protein, and lactose in breast milk. This study also confirmed the occurrence of a diurnal rhythm—changes in the protein and lactose content in human milk were observed within 12 h [2].

It is now also known that human milk is a reservoir of many microorganisms, and the theory of human milk as a sterile fluid has been rejected [3]. Human milk is a habitat of its own unique microbiome with beneficial, commensal, and potentially probiotic bacteria [4,5]. Assessment of the breast milk microbiome is carried out using classical microbiological methods (based on culture and identification) and increasingly using modern techniques, including sequencing [6]. Both approaches have limitations, which may be due to different sampling protocols and sample preservation (e.g., aseptic methods, time of day, and pre- or post-feeding sampling) and, at a later stage, DNA extraction or the selection of specific ones [6,7]. So far, it has been shown that the basic composition of the human milk microbiome includes genera such as *Staphylococcus*, *Streptococcus*, *Ralstonia*, *Pseudomonas*, *Serratia*, *Corynebacterium*, *Sphingomonas*, *Cutibacterium*, and *Bradyrhizobium*. It should be remembered that the number of microorganisms in HM is variable and depends on many factors [8,9,10,11] and that the correct HM microbiome also includes sequences related to protozoa, fungi, and viruses [12]. So, where does the human milk microbiome come from? Until recently, it was thought that every bacterial cell present in human milk is the result of contamination of the mother’s skin or the child’s oral cavity. In fact, only the detection of living bacterial cells or DNA of anaerobic species in human milk, which cannot survive in aerobic conditions and are usually in the intestinal environment, initiated discussion among researchers about the hypothetical sources of bacteria in human milk [13]. The literature indicates that the human milk microbiome shares common features with the microbiome of the child’s oral cavity and the gastrointestinal tract of the breastfeeding mother (Figure 1).

The reverse flow of breast milk from the infant’s oral cavity to the milk ducts suggests that microorganisms from the infant’s oral cavity may have an impact on the bacterial microbiome of breast milk. The presence of bacteria characteristics of the oral environment *Streptococcus salivarius*, *Streptococcus mitis*, *Rothia mucilaginosa*, and *Gemella* spp. in breast milk confirms this hypothesis [14]. This is also reflected in research on prematurely born children conducted by Biagi et al. (2018). These children initially received breast milk from a bottle. Only after the child’s general condition and normal sucking function had stabilized did they begin feeding children directly from the breast. It was then observed that after switching from bottle-feeding with expressed human milk to direct breastfeeding, the human milk microbiome became more diverse and dominated by typical oral microorganisms, i.e., *Streptococcus* and *Rothia* spp., confirming that contact with the infant’s oral cavity shapes the human milk microbiome [15].

Human milk contains many bacteria of the *Staphylococcus* genus, including the commensal bacteria characteristics of human skin: *S. epidermidis*, *S. hominis*, *S. haemolyticus*, and *S. lugdunensis*. Jimenez et al. (2008) showed that *S. epidermidis* was dominant in both human milk and the feces of breastfed infants. In contrast, in infants fed formula, *S. epidermidis* was less prevalent, which was a differentiating feature of both study groups [16]. Human skin commensal bacteria such as *Cutibacterium acnes* and species of the genus *Corynebacterium* are also frequently identified in human milk. In addition, the human skin commensal *Malassezia* spp. is the main genus of fungi present not only in human milk but also in and around the sebaceous glands [17]. Therefore, it can be assumed that the bacterial skin microbiota may colonize the mammary gland by entering through the nipple.

The literature on the subject indicates that the microbiome of human milk shares common features with the microbiome of the digestive tract of a breastfeeding mother. The genus *Saccharomyces*, which includes some of the most abundant fungi identified in the digestive tract, is also one of the main types of fungi present in human milk [18]. Human milk also contains the following types of bacteria: *Bifidobacterium*, *Veillonella*, *Bacteroides*, *Parabacteroides*, and *Clostridium* [19,20]. These bacteria are anaerobes that would not survive in the oxygen conditions of the child’s oral cavity and on the mother’s skin. Hence, the hypothetical “intestinal–mammary route”, which would explain the presence of the previously mentioned bacteria in human milk. Dendritic cells take up live bacteria by violating the intestinal epithelium. Then, in the mesenteric lymph nodes, these cells sequester live bacteria for several days, which are transported through the lymphatic system to specific parts of the body, including the mammary gland [14].

Certainly, HM colonization with microorganisms is a complex, dynamic, and not fully understood process, and the composition of the HM microbiome is modulated by maternal, environmental–cultural, and perinatal factors. Despite the growing interest among scientists in the human milk microbiome, there are not enough new studies assessing the microbiological composition of human milk.

Breast milk contains bacteria, both the “core” microbiome, which plays a positive role in the development of the child, as well as potentially pathogenic bacteria. Pathogenic bacteria can be a source of microorganisms constituting the permanent skin microbiota and transient microbiota resulting from contact with the environment. Therefore, the aim of the study was to assess the occurrence of bacteria in breast milk and nipple swabs before and after breastfeeding. The presence of bacteria cultured in human milk from Polish women giving natural births was assessed. Only a group of women with a normal BMI, originating from Poland, and giving birth at term vaginally were included in the study. In addition, the microbiological status of the areola and skin of the nipple before and after breastfeeding was assessed.

## 2. Materials and Methods

### 2.1. Study Group

The study was conducted in a group of 86 women living in Poland. Women with normal BMI and giving birth at term via vaginal delivery qualified for the study. The study participants voluntarily consented to participate in the study and completed questionnaires containing data such as age, BMI, HBD, type of delivery, place of residence, and education. Participants were recruited via social media.

The research was approved by the Bioethics Committee of the Nicolaus Copernicus University in Toruń at the Ludwik Rydygier Collegium Medicum in Bydgoszcz (consent no. KB121/2019).

### 2.2. Materials

The material for the study consisted of 86 human milk samples and swabs of the areola and nipple, before and after breastfeeding. The swabs were taken on swab sticks with transport medium (Profilab, Poland). The material (areola swab) was transported to the laboratory within an hour of collection (room temperature).

Human milk samples were collected in sterile containers and came from a daily collection (40 mL). During the day, the studied group of women expressed milk in four time slots: 06:00–12:00, 12:00–18:00, 18:00–24:00, and 24:00–6:00. The amount of expressed milk in each time interval was 10 mL, with 5 mL before the baby latches on and 5 mL after the end of feeding. Participants used a sterile breast pump to express milk at home. Human milk was stored at 2–4 °C. Human milk was analyzed within one hour of the end of daily collection.

### 2.3. Methods

#### 2.3.1. Assessment of the Presence of Bacteria (Culture-Based Method)

Breast swabs were inoculated (streak plate technique) onto Columbia Agar with 5% addition of Sheep Blood (Graso, Owidz, Poland). The cultures were incubated for 48 h at 37 °C, considering aerobic, absolutely anaerobic, and CO_2_-enriched atmosphere conditions.

A milk sample (100 µL) was inoculated (by surface culture method) onto CAB medium (incubated 48 h, 37 °C considering aerobic, absolutely anaerobic, and CO_2_-enriched atmosphere conditions) and MRS agar (De Mana–Rogosy–Sharpe Agar, Graso, Gdansk, Polnad) (incubation: 35 °C, 72 h, CO_2_).

#### 2.3.2. Species Identification

The cultured colonies were visually evaluated and based on the characteristic features of colony morphology, passage was carried out on CAB medium, and incubation was carried out under appropriate atmospheric conditions.

The obtained monocultures were subjected to species identification based on matrix-assisted laser desorption/ionization time-of-flight mass spectrometry (MALDI-TOF MS) technique, according to the manufacturer’s instructions. Acquisition and analysis of mass spectra were performed with a Microflex LT/SH mass spectrometer (Bruker, Bremen, Germany) using the MALDI Biotyper software package with the Bruker Taxonomy reference database (Bruker) and default parameter settings. The correctness of the identification performed with the MALDI Biotyper system is expressed in the form of a score index. A range of 2000 to 3000 (reliable identification of a microorganism to a species with a high level of certainty) was considered correct results of the analysis performed. The Bruker bacterial test standard (BTS; Bruker, which is an extract of *Escherichia coli* strain DH5 alpha reflecting a characteristic protein profile) was used for validation, according to the manufacturer’s instructions.

#### 2.3.3. Statistical Analysis

Statistical analysis was performed using the Statistica 13.1 software package from StatSoft^®^ (Kraków, Poland). The normality of the schedule was verified by the Shapiro–Wilk test. The distribution of the analyzed quantitative variables was found to be normal. Student’s *t*-test was used to assess statistical significance in two groups of independent variables. The variability of the parameters is presented in the form of a mean and standard deviation. The chi-squared test was used to compare the proportions between groups. A *p*-value less than 0.05 was considered significant.

Pearson’s correlations were used to examine correlations between variables. Correlations with *p* < 0.050 were considered statistically significant. Correlations were presented using heatmaps. Cluster analysis was tested using Ward’s method, and the Euclidean distance was adopted as a measure of distance. Hierarchical clustering analysis, presenting relationships and similarities between the type of bacteria and the studied material, was presented using a dendrogram and heatmaps.

## 3. Results

The evaluation of the presence of bacteria in breast milk (culture-based method) was characterized from 86 Polish women aged 31.25 ± 3.54 years, with normal BMI, giving birth at term via vaginal delivery. A significant number of women lived in the city (60%), had higher education (89.5%), and were primiparous (58%). Study participants were omnivores, did not take antibiotics during lactation or delivery, did not take medications during lactation, and did not have any chronic diseases. Women were supplemented with Omega 3 fatty acids (DHA—800 mg, EPA—68 mg), vitamin D (2000 IU), and folic acid (200 µg). Detailed characteristics of the women are presented in Table 1.

The analysis of human milk and areola and nipple swabs before and after breastfeeding, using culture methods, showed a diversity of bacterial species. Bacterial species from the genera *Streptococcus*, *Staphylococcus*, *Enterococcus*, *Micrococcus*, *Gemella*, *Dermacoccus*, *Neisseria*, *Veillonella*, *Corynebacterium*, *Rothia*, *Kocuria*, *Actinomyces*, *Pseudomonas*, *Enterobacter*, *Cutibacterium*, *Moraxella*, *Klebsiella*, *Escherichia*, *Acinetobacter*, *Pantonea*, *Bacillus*, *Limosilactobacillus*, *Schaalia*, *Leclercia*, *Cytobacillus*, *Lacticaseibacillus*, *Microbacterium*, *Psychrobacter*, *Shewanella*, *Stenotrophomonas*, *Raoultella*, *Aeromonas*, *Serratia*, and *Buttiauxella* were identified in human milk. Bacteria of lower diversity were isolated from areola swabs before breastfeeding. The following genera were identified: *Streptococcus*, *Staphylococcus*, *Enterococcus*, *Micrococcus*, *Gemella*, *Dermacoccus*, *Neisseria*, *Veillonella*, *Corynebacterium*, *Rothia*, *Kocuria*, *Actinomyces*, *Pseudomonas*, *Cutibacterium*, *Moraxella*, *Klebsiella*, *Escherichia*, *Acinetobacter*, *Pantonea*, *Bacillus*, *Limosilactobacillus*, *Schaalia*, *Leclercia*, *Cytobacillus*, and *Lacticaseibacillus*. The lowest diversity was obtained from the areola swab after breastfeeding (genera: *Streptococcus*, *Staphylococcus*, *Enterococcus*, *Micrococcus*, *Gemella*, *Dermacoccus*, *Neisseria*, *Veillonella*, *Corynebacterium*, *Rothia*, *Kocuria*, *Actinomyces*, *Pseudomonas*, *Enterobacter*, *Cutibacterium*, *Moraxella*, *Klebsiella*, *Escherichia*, *Acinetobacter*, *Pantonea*, *Bacillus*, and *Limosilactobacillus*). A detailed list of isolated bacterial species is presented in Table 2. The percentage share of isolated species in the tested material is presented in the pie charts in Figure 2, Figure 3 and Figure 4. A comparison of the percentages of isolated species from each tested biological material is presented in Figure 5.

A total of 120 different bacterial species were isolated from 86 women, including 107 strains from breast milk, 62 strains from areola swabs before breastfeeding, and 50 strains from areola swabs after breastfeeding. Several species (N = 43)—*M. paraoxydans*, *P. sanguinis*, *S. oneidensis*, *S. maltophila*, *R. ornithinolytica*, *A. caviae*, *S. marcescens*, *B. gaviniae*, *L. gasseri*, *L. oris*, *L. crispatus*, *L. brevis*, *L. salivarius*, *L. rhamnosus*, *B. breve*, *E. brevis*, *S. infantis*, *S. gollolyticus*, *S. saprophyticus*, *S. borealis*, *E. italicus*, *E. durans*, *A. parvus*, *A. baumannii*, *P. agglomerans*, *P. septica*, *L. reuteri*, *L. fermentum*, *C. horneckiae*, *C. acnes*, *C. lipophiloflavum*, *C. accolens*, *P. putida*, *P. koreensis*, *P. cedrina*, *P. oleovorans*, *E. hormaechei*, *E. bugandensis*, *E. ludwigii*, *E. roggenkampii*, and *E. asburiae*—were isolated exclusively from human milk. On the other hand, unique species from the areola after breastfeeding were *P. anthophila*, *M. catarrhalis*, *C. pseudodiphtheriticum*, *C. amycolatum*, and *C. propinquum*. The Venn diagram (Figure 6) shows the relationship between the isolated bacterial species from human milk and areola swabs. In all the biological materials examined, we observed 33 genera mainly from *Streptococcus*, *Staphylococcus*, *Neisseria*, *Corynebacterium*, *Rothia*, *Pseudomonas*, *Acinetobacter*, and *Micrococcus*. The comparison of the occurrence of bacteria in the studied groups is presented in Table 3.

Cluster analysis presented as a dendrogram allowed for grouping the types of bacteria in terms of the number of species isolated from breast milk and areola swabs before and after breastfeeding the child (Figure 7). It was shown that *Streptococcus* spp. and *Staphylococcus* spp. form a separate cluster, which indicates a distribution that is definitely different from the other clusters. These clusters show the microbial core of human milk. Additionally, heatmaps comparing the numbers of species in the tested biological materials show the highest numbers of *Streptococcus* spp. and *Staphylococcus* spp. in the tested material. It was also shown that *Lacticaseibacillus* spp. and *Bifidobacterium* spp. form the most similar cluster, as do *Escherichia* spp. with *Cutibacterium* spp. and *Moraxella* spp. with *Pantonea* spp. However, the heatmaps show the lowest numbers in the abovementioned subgroups.

The analysis showed positive and negative correlations between the types of bacteria in the studied groups (Figure 8A–C).

An increase in the number of *Streptococcus* species was observed along with the increase in the genera *Neisseria* (r = 0.344, *p* = 0.023), *Rothia* (r = 0.399, *p* = 0.001), *Kocuria* (r = 0.224, *p* = 0.027), and *Actinomyces* (0.355, *p* = 0.003) in the swab before breastfeeding. A negative correlation was noted between *Staphylococcus* spp. and *Rothia* spp. (r = −0.226, *p* = 0.024), and *Staphylococcus* spp. and *Klebsiella* spp. (r = −0.237, *p* = 0.005), and a positive correlation between *Enterococcus* spp. and *Escherichia* spp. (r = 0.491, *p* = 0.001), *Dermacoccus* spp. and *Moraxella* spp. (r = 0.445, *p* = 0.001), and *Neisseria* spp. and *Veillonella* spp. (r = 0.508, *p* = 0.001).

In the swab taken after breastfeeding, a decrease in the number of *Streptococcus* species was observed along with an increase in *Enterococcus* spp. (r = −0.263, *p* = 0.023) and *Bacillus* spp. (r = −0.218). A negative correlation was also observed between *Streptococcus* spp. and *Neisseria spp.* (r = −0.232, *p* = 0.028). Positive correlations were observed between *Enterococcus* spp. and *Bacillus* spp. (r = 0.417, *p* = 0.001), *Enterococcus* spp. and *Pantonea* spp. (r = 0.491, *p* = 0.001), *Micrococcus* spp. and *Moraxella* spp. (r = 0.278, *p* = 0.015), *Moraxella* spp. and *Acinetobacter* spp. (r = 0.494, *p* = 0.001), and *Escherichia* spp. and *Pantonea* spp. (r = 0.571, *p* < 0.001).

In human milk, an increase in *Streptococcus* spp. species was observed along with an increase in *Rothia* spp. (r = 0.306, *p* = 0.001) and *Kocuria* spp. (r = 0.254, *p* = 0.023). Positive correlations were observed between *Enterococcus* spp. and *Escherichia* spp. (r = 0.347, *p* = 0.001), *Micrococcus* spp. and *Bifidobacterium* spp. (r = 0.266, *p* = 0.020), *Dermacoccus* spp. and *Limosilactobacillus* spp. (r = 0.349, *p* = 0.001), *Neisseria* spp. and *Actinomyces* spp. (r = 0.391, *p* = 0.001), *Micrococcus* spp. and *Corynebacterium* spp. (r = 0.312, *p* = 0.005), *Kocuria* spp. and *Moraxella* spp. (r = 0.317, *p* = 0.001), and *Pseudomonas* spp. and *Enterobacter* spp. (r = 0.366, *p* = 0.001).

## 4. Discussion

Breast milk is constantly being studied scientifically. Experts are learning more and more about the composition and benefits of breast milk. HMM (human milk microbiome) is an important factor in the colonization of the digestive tract in children, which is why it is crucial to avoid any form of disturbance in HMM that can change the microbiological balance, especially in the first 100 days of a child’s life. Microbiological dysbiosis can be a trigger for the development of necrotizing enterocolitis, especially in premature infants, as well as the onset of chronic diseases such as asthma and obesity later in life [21]. The composition of the human milk microbiome has a dual impact on the infant’s health: promoting immunological homeostasis in the gut and facilitating digestive processes [22]. Hufnagl et al. emphasize that bacterial diversity plays a key role in maintaining immunological balance in both infants and adults [23]. Specifically, bacteria derived from human milk transmit early antigenic stimuli that have a positive effect on the maturation of the intestinal immune system in infants [22]. Scientific evidence indicates that approximately 25–30% of the intestinal microbiota in infants originates from human milk [24].

In this study, the presence of breast milk bacteria (culture-based method) of Polish women giving birth at term via vaginal delivery was evaluated. The use of only culture-based methods has some limitations, as it only allows data to be collected on culturable species. However, these methods allow the identification of low abundance species that are not detectable by metagenomics [25]. With the development of science, more and more modern molecular biology techniques, including sequencing, are being used to characterize breast milk [6]. The use of new techniques has made it possible to confirm the rich and diverse microbial community (including non-culture method) in human milk samples [6,7]. It should be emphasized that non-culture methods also have disadvantages. According to Douglas et al. (2020), the method of DNA extraction has a significant impact on the obtained results of microbiome evaluation. The researchers [26] emphasize the importance of using rigorous, well-validated DNA extraction methodologies and conservatively interpreting the data obtained, especially for low-biomass samples. A comparative study by Lee et al. (2024) showed that sequencing provided data on more genera in breast milk than culture-based methods [27]. Only culture methods and a limited number of media were used in this study. However, breast swabs were taken before and after feeding to obtain more data on potential microbial transmission routes and milk composition. Despite this limitation, this pilot study provided valuable data on culturable bacterial species among patients in Poland.

### 4.1. Core Microbiota of Human Milk

Based on the results of culture methods, we have shown that the most frequently detected genera were *Staphylococcus* (20%) and *Streptococcus* (19%), which is consistent with other previous research findings [9,28,29,30,31]. Scientists suggest that both *Staphylococcus* spp. and *Streptococcus* spp. are two bacterial groups that constitute the “core” composition of the human milk microbiota, the occurrence of which is independent of geographical location and the analytical technique used [9,28,31]. It was shown that *S. epidermidis* and *S. hominis* were the predominant staphylococci present in the examined human milk, which aligns with the results obtained by Gonzalez et al. [30]. Both species are recognized as common skin bacteria and bacteria of a healthy maternal milk environment [32]. In our study group, there were women who delivered at term; however, interestingly, Soeorg et al. observed a higher abundance of *S. hominis* and *S. epidermidis* in preterm milk, which was most likely due to the fact that non-virulent strains of these species are early intestinal pioneers in term infants and may have the potential to reduce colonization by more virulent species [33]. The presence of these staphylococci was also confirmed in swabs taken before and after breastfeeding. *S. lugdunesis* was also isolated from the milk samples tested. Isolation of this species from milk was also demonstrated by Chen et al. (2016) and Miura et al. (2023) [34,35]. In the microbiological swab taken before feeding and in the human milk sample, the presence of *S. petrasii* was confirmed Scientific reports do not indicate that human milk is a common habitat for this bacterium. In a study conducted by Xie et al. [36], *S. petrasii* was the dominant colostrum species of Han tribe women. These results indicate species variation by latitude.

The occurrence of species within the genus *Streptococcus* identified in our study confirms previous studies [30,31]. The few available studies that presented results at the species level indicate *S. mitis* and *S. salivarius* as part of the dominant species in human milk, constituting part of the “core” microbiota of human milk [6,37,38,39]. *Streptococcus* spp. are bacteria commonly found in the oral cavity of infants [39], and their presence in human milk is explained by the backflow of milk from the infant’s mouth back into the mammary gland during breastfeeding [40]. In our study, we observed that *S. mitis* dominated in microbiological swabs taken after breastfeeding. The presence of *S. mitis* in swabs after breastfeeding and in human milk may indicate a child’s oral cavity route. *S. parasanguinis* dominated in the microbiological swab before breastfeeding and in human milk samples. This bacterium colonizes many sites in the body [41]. It is commonly found in the oral cavity of both children and adults [42,43], but it is also the dominant bacterial species in the small intestine of adults [44]. Considering the presence of *S. parasanguinis* in human milk, the intestinal–mammary route should be considered as a hypothetical source of the human milk microbiome. It is also interesting that *S. parasanguinis* is one of the dominant, pioneering bacteria colonizing the infant’s gut in the first days of life [45]. According to available data [46], *S. salivarius* and *S. parasanguinis* (isolated from the milk samples in this study), have probiotic potential and may enhance host immune tolerance in early life and resistance to inflammatory pathologies in adulthood. These reports are consistent with the results of Damaceno et al., who also emphasize the probiotic potential of *S. salivarius* strains present in human milk in their studies [47].

In our study, the third most abundant type of bacteria in human milk was *Acinetobacter* (7%). It should be noted that in the microbiological swabs taken both before breastfeeding and after breastfeeding, the percentage of this bacterium was much lower and amounted to 2% and 1%, respectively. *Acinetobacter* spp. is considered the microbiological “core” of human milk, which is controversial for some scientists [10]. On the other hand, other scientists have detected *Acinetobacter* spp. in human milk samples, in which milk collection, as in our studies, was carried out without aseptic cleaning of the breasts, as well as with the rejection of the first milk [48,49]. Sakwinska et al., however, concluded that it is unlikely that the predominance of *Acinetobacter* spp. resulted from the human milk collection protocol and believe that *Acinetobacter* spp. may be a specific feature of the microbiota associated with breastfeeding [48]. It should be noted that *Acinetobacter* spp. is a bacterium commonly found in soil [50]; therefore, other researchers have linked the presence of *Acinetobacter* spp. in human milk with a maternal diet based on legumes [51] or with the proximity of the soil environment [52].

In our study, we isolated a high number of *M. luteus* (21%) and *E. faecalis* (21%) strains in a microbiological swab from the nipple and areola skin before breastfeeding. These results support the hypothesis that the source of the breast milk microbiome are bacteria residing on the mother’s skin. *E. faecalis* is also often found in persistent, non-healing wounds, but its contribution to chronic wounds remains insufficiently studied [53]. According to the latest systematic review, *Enterococcus* spp. is one of the most frequently identified bacteria in breast milk [54]. It should also be noted that the presence of the *Enterococcus* genus in human milk partially influences the composition of the infant’s gut microbiota, with some strains having a key effect. Anjum et al. showed that the *E. faecalis* NPL-493 strain is the most promising candidate for a probiotic, showing significant tolerance to acid, bile, and digestive enzymes in the human gastrointestinal tract and antibacterial activity against many pathogens. The study showed that *E. faecalis* NPL-493 from human milk is safe and can be considered a potential probiotic [55]. Interesting observations regarding the presence of *Enterococcus* bacteria are presented by Laursen et al., who observed that the lack of enterococci from mother’s milk inhabiting the intestines may contribute to excessive weight gain in breastfed infants [56].

The human milk microbiome is diverse, and among the dominant genera, *Enterobacter* spp. is also mentioned [5,36,40], which was also isolated in this study. It should be noted that in our study, *Enterobacter* spp. was isolated only from human milk (6%) and in a microbiological swab after breastfeeding (1%). The presence of *Enterobacter* spp. in human milk confirms the evidence for the existence of an “intestinal–mammary route” that allows the transfer of bacteria from the maternal intestines to the milk ducts [57].

In our study, we also isolated bacteria of the genus *Rothia* in human milk. According to the latest systematic review, *Rothia* spp. is among the ten most frequently identified bacteria in human milk. The range of relative abundance of *Rothia* spp. in the literature ranges from 1% to 6% [42], which is consistent with the abundance of this bacterium observed in our study (5%). *Rothia* is commonly found in human saliva. This fact can also be observed based on our study, as the relative abundance of *Rothia* spp. in the swab from the skin of the areola and nipple before breastfeeding was 5%, while in the swab after breastfeeding it was 10%. Another bacterium commonly found in the oral cavity and saliva of infants [58], which was isolated in our study from human milk, was *Neisseria* spp. (4%). In pre- and post-breastfeeding swabs, strains of this genus were detected in 1% and 6% of samples, respectively. Despite the fact that Neisseria spp. is commonly found in the oral cavity, it is not a basic type of human milk bacteria, which may indicate a child’s oral cavity route [59].

In our study, several species characterized as potentially beneficial to the health of infants were also isolated. The genera *Lactobacillus* and *Bifidobacterium* are most often used as probiotic cultures [60]. *L. gasseri* isolated from our human milk samples shows good survival in the gastrointestinal tract and is associated with many probiotic activities and roles, including the reduction in mutagenic enzymes in the feces, the production of bacteriocins, and the stimulation of immunomodulation of macrophages. Scientific reports indicate that both *L. gasseri* as well as *L. salivarius* present in our human milk samples show positive results in the treatment of mastitis and also reduce the risk of recurrence of mastitis [61]. From the tested human milk samples, the following were also isolated: *L. oris*, *L. brevis*, *L. rhamnosus*, as well as *L. crispatus*. The isolation of the *L. crispatus* strain in our study is considered groundbreaking because, so far, the literature indicates the identification in human milk of such species as *L. casei*, *L. fermentum*, *L. gasseri*, *L. gastricus*, *L. plantarum*, *L. reuteri*, *L. rhamnosus*, *L. salivarius*, and *L. vaginalis* [62]. *L. crispatus* is the dominant species of the vaginal microbiome of most women of reproductive age [63]. It should be noted that our study group consisted of women giving birth exclusively vaginally, and it has been proven that newborns born naturally acquire bacteria from the mother’s birth canal [64]. In our study, the incidence of bacteria of the genus *Lactobacillus* was 4%, which is consistent with the relative incidence of this bacterium found by other researchers (1–5%) [54]. The literature indicates a lower incidence of bacteria of the genus *Lactobacillus* in milk samples of women undergoing antibiotic therapy or giving birth by cesarean section [62]. A relationship was also found between the reduced detection of *L. salivarius* and the use of anesthesia during childbirth [65]. Bacterial species of the genus *Limosilactobacillus*—*L. reuteri*, *L. fermentum*, and *L. brevis*—which were isolated from human milk, also have probiotic properties. Interestingly, studies suggest that the presence of *L. fermentum* CECT 5716 may be particularly important for children born by cesarean section because this population is exposed to a higher risk of infection [66]. Another species of bacteria, which was isolated in our human milk samples with probiotic properties, is *B. breve*. The isolation of the *B. breve* strains in human milk is another proof of the presence of the “intestinal–mammary route”. Kordy et al. identified *B. breve* in the mother’s rectum, mother’s milk, and in the infant’s intestines. The results of this study confirm the presence of the intestinal–mammary tract, a reverse mechanism of milk inoculation, which enables the transfer of bacteria, such as *B. breve*, from the mother’s intestines to the milk ducts through bacterial translocation via immune system cells [67].

The “core” of the human milk microbiome also includes *Corynebacterium*, a commensal bacterium of human skin. The presence of this bacterium was observed in the feces of mothers, as well as in the intestines and feces of infants, which indicates that *Corynebacterium* spp. colonizes the intestines of infants through human milk [58]. It should also be noted that Corynebacterium produces not only amino acids such as glutamic acid and lysine, but also nucleotides and vitamins, which may have a direct impact on the multiplication of the mentioned organic compounds in breast milk [68].

Another important bacterium that is part of the basic microbiome of human milk was *Veillonella* spp. According to the literature, the range of incidence of *Veillonella* spp. in human milk is <1% to 6%. In our human milk samples, the incidence of this bacterium was 1%. *Veillonella* spp. in human milk appears only during direct breastfeeding, which confirms that this bacterium comes from the child’s oral cavity [69]. Hence, in the microbiological swab from the skin of the nipple and areola after breastfeeding, which we analyzed, the incidence of *Veillonella* spp. was higher than in human milk samples. The isolation in human milk of bacteria of the genus *Gemella* spp., like *Veillonella* spp., is associated with the child’s oral cavity and especially with the upper respiratory tract [10]. In our study, the incidence of *Gemella* spp. in human milk was 1%. However, the available literature indicates that the range of relative incidence of *Gemella* spp. ranges from 0.7 to 13%. It is possible that such a large range of relative incidence is related to the fact that carriage of *Gemella* spp. has been linked to allergies and asthma in children [43].

The genus *Pseudomonas* (3%), isolated from our human milk samples, is commonly identified in human milk by other scientists [54].

### 4.2. Potential Pathogens and Contamination Risks

In addition to the genera constituting the core microbiome, pathogenic species were found in the tested milk and swabs before and after breastfeeding. The source of these microorganisms is likely to be the environment. *S. haemolyticus* was isolated from the tested samples. *S. haemolyticus* can be an etiological agent of opportunistic infections in hospitalized and immunocompromised patients [70]. The deliveries of all patients took place in a hospital setting. The hands of medical staff can be a vector for *S. haemolyticus*, which presumably could have been the source of this staphylococcus in the samples analyzed. *S. borealis* and *S. saprophyticus* were also isolated from the tested milk samples. *S. borealis* is isolated from cow’s milk and can contribute to udder infections in cattle, while *S. saprophyticus* is an etiological factor in urinary tract infections in young women [71]. Another staphylococcus isolated from milk samples was *S. aureus*. This species may be the etiological agent of mastitis among lactating women [72]. Patients from whom milk samples were collected did not have active breast inflammation in the form of breast inflammation, breast swelling, or milk stasis. However, observations by Heikkilä et al. demonstrate that isolates of *S. epidermidis* and *S. salivarius* exhibit the ability to inhibit the *S. aureus* infection [32]. These same properties are shown by *C. acnes* (formerly *Propionibacterium acnes*), which, by secreting by-products of glycerol fermentation, inhibits the growth of *S. aureus* [73]. According to Hunt et al. (2012), human milk oligosaccharides (HMOs) can promote the growth of staphylococci in breast milk, which confirms the results we obtained related to the isolation of 11 species of the genus *Staphylococcus* [7].

The presence of *A. baumannii* was confirmed in the milk sample. *A. baumannii* is an opportunistic pathogen that causes hospital-acquired infections. In recent years, an increase in multidrug-resistant strains of *A. baumannii* has been noted, which affects therapeutic difficulties [74]. However, research by Spicer et al. (2021) and Ackerman et al. (2018) indicates that breast milk HMOs inhibit the growth of *A. baumannii* [75,76].

Disturbing in our study was the numerous presence of *Bacillus cereus* (6%) in human milk. The presence of *B. cereus* was also confirmed in swabs before and after breastfeeding. *B. cereus* may be an etiological factor in food poisoning that manifests itself as a vomiting or diarrheal syndrome (toxin-producing strains) [77]. *B. cereus* can be a factor in severe infections in premature infants [78]. Importantly, *B. cereus* is a major threat in human milk banks, and a high rejection rate of samples due to contamination with this microorganism is recorded [79]. Nevertheless, the presence of *B. cereus* in human milk is quite disturbing, especially since studies conducted in the same geographical area (Poland) in the years 2013–2021 did not show the presence of *B. cereus* [31]. *B. cereus* is present in hospital and natural environments [80], so it seems likely that airborne, direct, or indirect contact could be the source of this microorganism in the milk samples and swabs tested before and after staining.

In the present study, the following were isolated from milk samples: *C. tuberculostearicum* and *C. kroppenstedtii. C. tuberculostearicum* was shown to be the etiologic agent of breast abscess. Identification of this species is difficult, and the MALDI TOF MS technique is applicable [81]. In contrast, *C. kroppenstedtii* was isolated from female patients and mainly from breast abscesses and cases of granulomatous mastitis [69]. Patients recruited in the present study showed no abnormalities or inflammation in the mammary glands. The present study indicates the isolation of *Corynebacterium* spp. that constitutes the natural microbiota of the skin as well as potentially pathogenic species.Genus of *Klebsiella* are also often identified in human milk [31,82]. *K. pneumoniae* and *K. oxytoca* were isolated before breastfeeding as well as from human milk, while in a microbiological swab from the skin of the areola and nipple after breastfeeding we identified *K. oxytoca. Klebsiella* spp. physiologically colonizes the intestines, although it can also be found on the skin and nasopharynx, and the appearance of this bacterium in human milk is probably conditioned by direct contact with the mother’s skin and the child’s oral cavity. *Klebsiella* spp. has also been isolated from soil and water [58]. It should be emphasized that *K. pneumoniae* is a pathogenic microorganism. A pre-survey by Pawlik et al. (2017) conducted in Poland shows that *K. pneumoniae* was the etiological agent of sepsis in newborns, and the source of these Gram-negative rods was breast milk [83]. Lopez Leyva et al. emphasize the potential of environmental factors to modify the human milk microbiome [58]. These observations may suggest that the breastfeeding women in our study may have acquired these bacteria from the environment or from the infant via the retrograde flow of bacteria from the oral cavity to the milk ducts of the mammary gland. Monitoring for potentially pathogenic bacteria in breast milk is an important aspect of this research.

Bacteria of the genus *Pseudomonas* in human milk are generally considered an environmental contaminant [12]. Species were also isolated from milk samples: *P. oleovorans* and *P. koreensis.* Interestingly, *P. koreensis* was first isolated from agricultural soils [84]. It was also found in animal milk (yak), where it presented inhibitory antimicrobial activity against Gram-positive and Gram-negative bacteria [85]. Similar conclusions are suggested by the presence of *E. coli*, which we isolated both in human milk and in a microbiological swab from the skin of the areola and nipple before and after breastfeeding. This relatively anaerobic bacterium is part of the physiological intestinal microbiota of the large intestine. Therefore, the presence of *E. coli* in human milk may be a confirmation of the presence of the “entero-mammary pathway”. However, it should be noted that *E. coli* can contaminate both food and water [86]; hence, there is a likely presence of this bacterium on the skin of the areola and nipple of the study participants.

In a pre-provided study, *Aeromonas caviae* was isolated from milk samples. Isolation of *A. caviae* from cow’s milk samples at the 13% level in Greece was demonstrated by Melas et al. (1999) [87]. *A. caviae* is the etiological agent of diarrhea in infants [88].

### 4.3. Geographical and Maternal Influences

In conclusion of the above manuscript, we would like to emphasize the strength of our study, which was the inclusion criteria of breastfeeding women in the study. We are aware that specific maternal factors, such as maternal body weight, mode of delivery, gestational age, and geographical location, modify the human milk microbiota. Therefore, we recruited 86 breastfeeding women with normal BMI, originating from Poland, and giving birth at term vaginally. Scientific literature shows that the HM microbiome is different in rural and urban women, and in women living in different countries and even cities. Taghizadeh et al., conducting a study in Iran, observed that the total number of *Lactobacillus* spp. was higher in the milk of mothers living in rural areas compared to mothers living in urban areas [89]. Further studies indicating that geographical location influences the composition of the HM microbiome show lower abundance of *Bifidobacterium*, *Propionibacterium*, *Veillonella*, and *Serratia* in the milk of women from the Central African Republic compared to the milk of women from Switzerland and the USA [6,90]. Lackey et al. conducted a large study covering 11 centers worldwide and observed that among women giving birth vaginally, the highest number of *Bacteroidetes* in milk was found in the milk of Spanish women compared to the milk of women from China, Finland, or South Africa. In turn, the milk of Chinese women who gave birth by cesarean section had the highest number of *Actinobacteria*, compared to the group of women giving birth by cesarean section and living in Europe and Africa. The results of this study are consistent with previous studies comparing milk samples from different places around the world—USA, Ethiopia, Kenya, Gambia, Peru, Sweden, and Spain—which show that the composition of HM differs significantly within and between cohorts [38]. Another very important factor differentiating the composition of the human milk microbiome is gestational age. In the milk of women giving birth prematurely, a higher number of *Enterococcus* spp. and a lower number of *Bifidobacterium* spp. are observed, compared to the milk of women who gave birth at term [91]. However, the most important factor for the diversity and colonization pattern of the intestinal microbiome in the first three months of the child’s life, according to a systematic review, is the mode of delivery [92]. This is also confirmed by studies conducted on 596 children who were born at term, and the most important factor influencing the composition of the intestinal microbiota of children was the mode of delivery [93]. Children who are born vaginally acquire bacteria from the mother’s birth canal, and these are mainly species of the genera *Lactobacillus* and *Prevotella.* Shao et al. found that cesarean sections reduce maternal bacteria transmission and increase colonization by hospital-associated species like *Enterococcus*, *Enterobacter*, and *Klebsiella*. This indicates that delivery mode significantly impacts infant gut microbiota composition from the neonatal period to infancy [93].

Babies born by cesarean section have lower levels of anaerobic bacteria, such as *Bacteroides* spp. and *Bifidobacterium* spp. Bacteria from the genera *Clostridium*, *Cutibacterium*, and *Corynebacterium* are also present [94]. An association has also been noted between cesarean delivery and the lower abundance and diversity of the phyla *Actinobacteria* and Bacteroidetes in the gut microbiota, and the higher abundance and diversity of the phylum *Firmicutes*, recorded from birth to 3 months of age [95]. The human milk microbiome is also dependent on maternal factors, including the occurrence of chronic diseases, maternal BMI, diet, use of probiotics and antibiotics, and previous infections [91]. Milk with lower microbiological diversity and significantly lower abundance of the *Bifidobacterium* genus—and higher abundance of the *Staphylococcus*, *Akkermansia*, and *Granulicatella* genera—is observed more often in overweight and obese women than in women with normal body mass (BMI). It has also been observed that *Staphylococcus*, *Corynebacterium*, and *Brevundimonas* occur much more often in the milk of obese women than in overweight women with normal BMI [96,97].

### 4.4. Limitations and Future Directions

Our pilot study presents an analysis of the bacterial composition of human milk, including pre- and post-breastfeeding swabs, using culture-based methods. Such studies with Polish women are rare, which underlines the importance of addressing this issue. However, the study has certain limitations. In this study, only culture methods were used, allowing only culturable bacteria to multiply. The methodology used and the use of blood agar and MRS did not assess the presence of difficult-to-cultivate or non-culturable bacteria. In addition, fungal media were not included in the study. The study was a pilot study and, in the future, we plan to extend the study with molecular biology methods, including sequencing.

## 5. Conclusions

The results we presented indicate the presence of many species of bacteria in breast milk. Importantly, the presence of lactic acid bacteria (LAB) including the genera *Lactobacillus*, *Bifidobacterium*, and *Limosilactobacillus* were also demonstrated. Thus, breastfeeding can be a valuable way of feeding newborns and infants. The results obtained certainly prove the importance of promoting exclusive and dominant breastfeeding in order to increase the diversity of bacteria in the child’s intestinal microbiota, which will have a positive impact on his health. The study also made it possible to identify bacteria in breast milk that come from the mother’s skin and intestines (entero-mammary route) as well as those that are characteristic of the child’s oral cavity. Further studies on a larger number of patients and the use of molecular biology methods are indicated.

## Figures and Tables

**Figure 1 biology-14-00332-f001:**
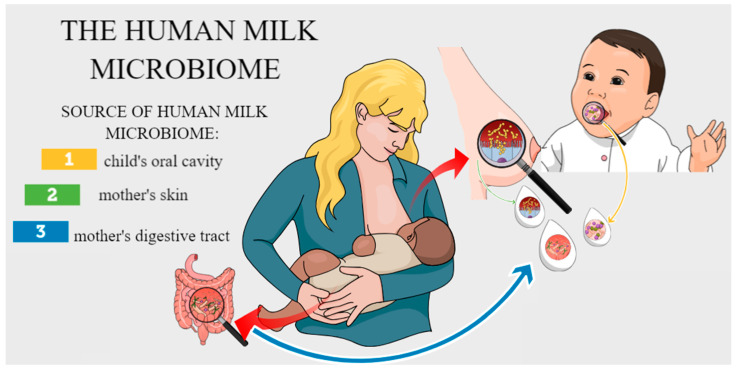
Hypothetical sources of the human milk microbiome.

**Figure 2 biology-14-00332-f002:**
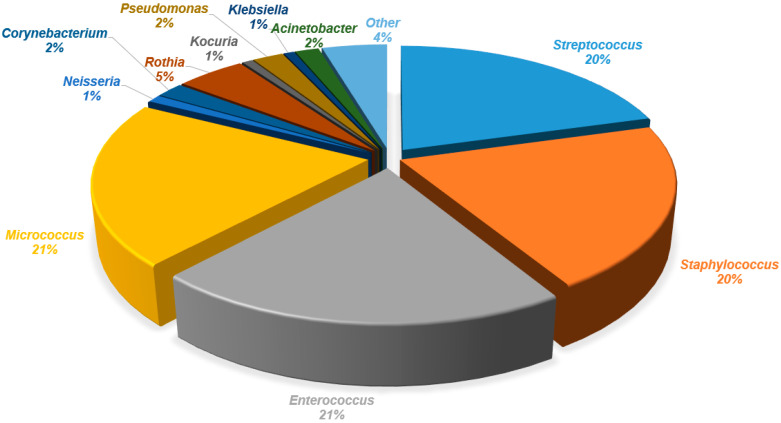
Percentage of bacterial genera isolated from the areola before breastfeeding. Other: *Lacticaseibacillus*, *Schaalia*, *Leclercia*, *Cytobacillus*, *Lacticaseibacillus*, *Bacillus*, *Moraxella*, and *Escherichia*.

**Figure 3 biology-14-00332-f003:**
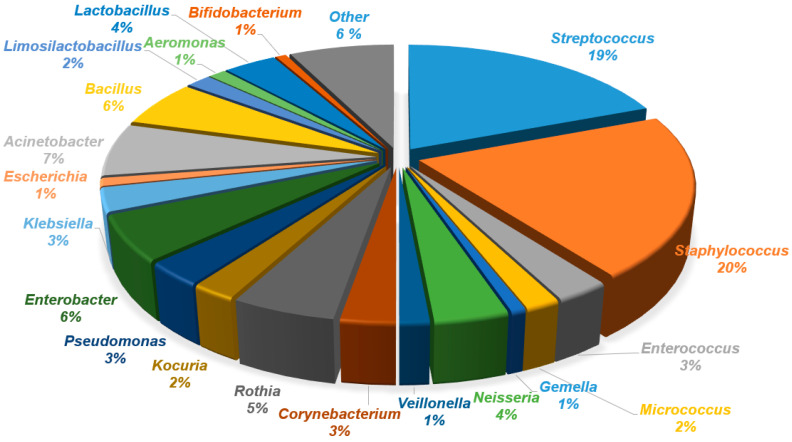
Percentage of bacterial genera isolated from human milk. Other: *Lacticaseibacillus*, *Bifidobacterium*, *Empeadobacter*, *Cytobacillus*, *Lacticaseibacillus*, *Microbacterium*, *Psychrobacter*, *Shewanella*, *Stenotrophomonas*, *Raoultella*, *Aeromonas*, *Serratia*, *Buttiauxella*, *Schaalia*, *Leclercia*, and *Pantonea*.

**Figure 4 biology-14-00332-f004:**
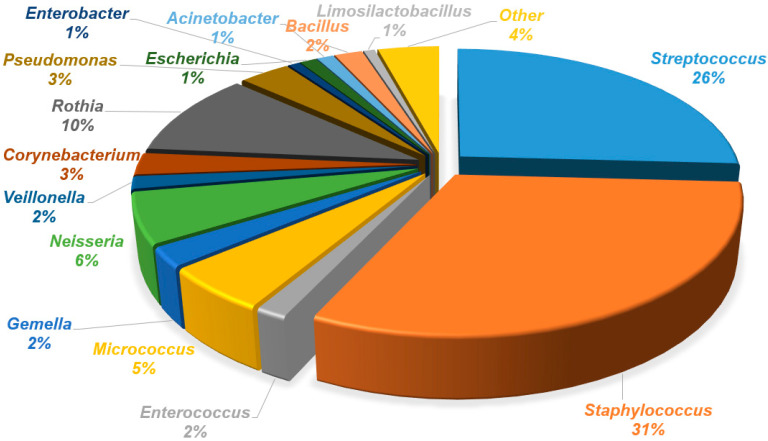
Percentage of bacterial genera isolated from the areola after breastfeeding. Other: *Lacticaseibacillus*, *Pantonea*, *Cutibacterium*, *Moraxella*, *Klebsiella*, and *Enterobacter*.

**Figure 5 biology-14-00332-f005:**
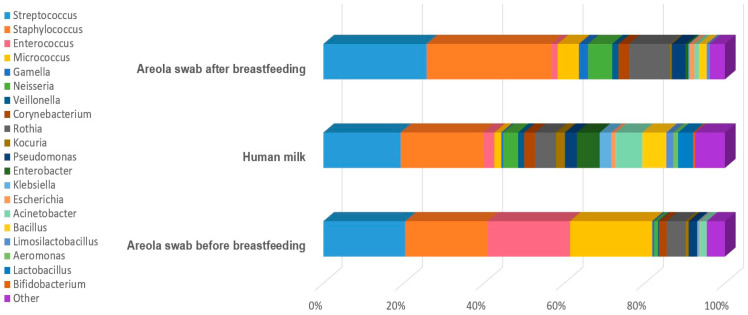
Comparison of isolated genera from: the areola before and after breastfeeding and from human milk. Other: *Lacticaseibacillus*, *Pantonea*, *Cutibacterium*, *Moraxella*, *Klebsiella*, *Enterobacter*, *Schaalia*, *Leclercia*, *Cytobacillus*, *Bacillus*, *Moraxella*, *Escherichia*, *Bifidobacterium*, *Empeadobacter*, *Microbacterium*, *Psychrobacter*, *Shewanella*, *Raoultella*, *Aeromonas*, *Serratia*, *Buttiauxella*, *Schaalia*, and *Leclercia*.

**Figure 6 biology-14-00332-f006:**
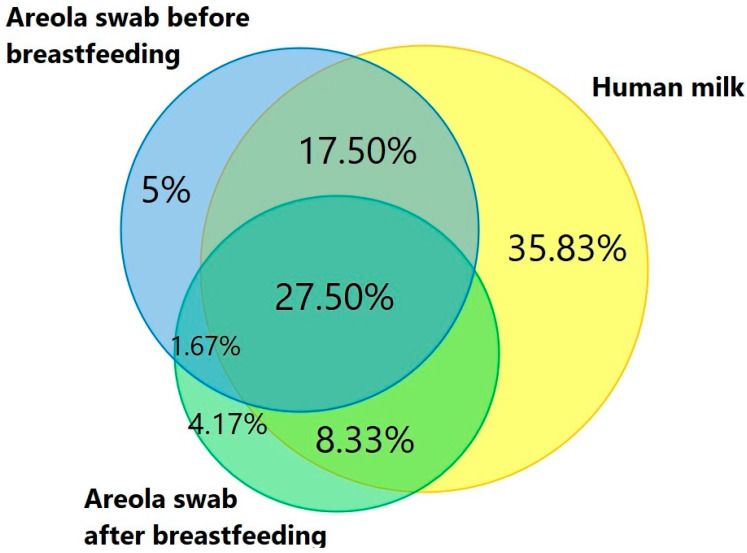
Venn diagram showing unique and shared bacterial strains between human milk and areola swab samples before and after breastfeeding.

**Figure 7 biology-14-00332-f007:**
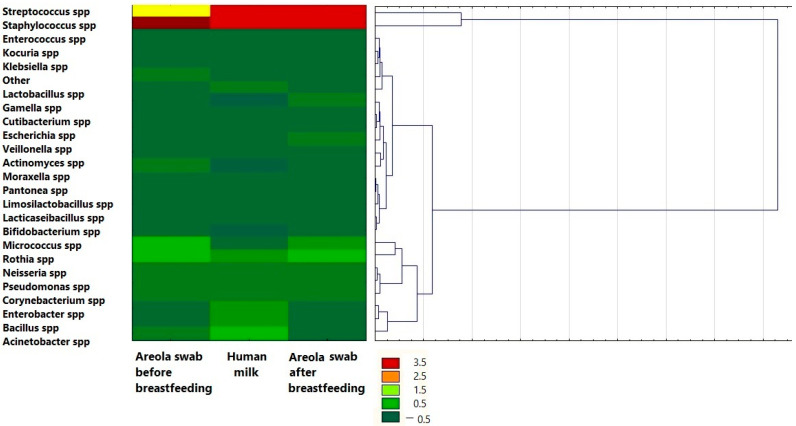
Dendrogram with heatmap showing common taxa of human milk and areola swab. Colormap from green to red shows increasing absolute abundance. Green shows lowest abundance and red the highest.

**Figure 8 biology-14-00332-f008:**
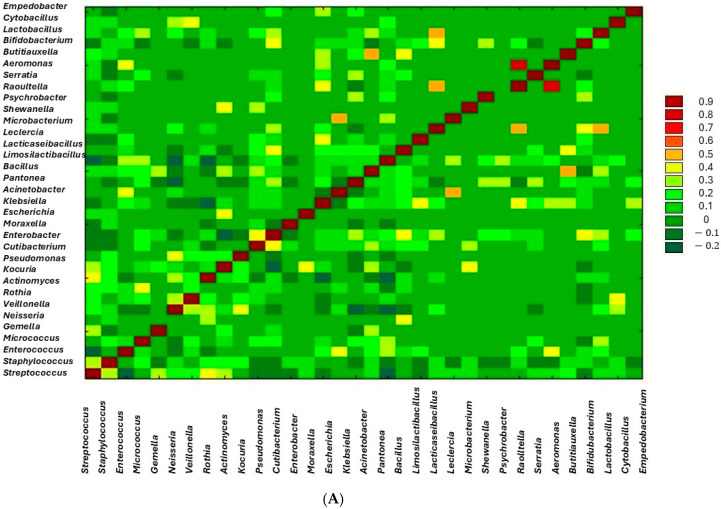
(**A**) Heatmap showing Pearson correlations between bacterial genera in the human milk. The legend shows the correlation coefficient values, while the axes show the genera of bacteria. (**B**) Heatmap showing Pearson correlations between bacterial genera in the areola swab before breastfeeding. The legend shows the correlation coefficient values, while the axes show the genera of bacteria. (**C**) Heatmap showing Pearson correlations between bacterial genera in the areola swab after breastfeeding. The legend shows the correlation coefficient values, while the axes show the genera of bacteria.

**Table 1 biology-14-00332-t001:** General characteristics of breastfeeding women (N = 86).

Variable	Mean ± SE
Age [year]	31.25 ± 3.54
BMI [kg/m^2^]	23.95 ± 4.18
HBD	39.39 ± 1.19
Place of residence	N (%)
City	52 (60%)
Village	34 (40%)
Education	N (%)
Secondary	9 (10.5%)
Higher	77 (89.5%)
Parity	N (%)
Primiaraus	50 (58%)
Multiparaus	36 (42%)
Number of delivery—2	31 (86%)
Number of delivery—3	5 (14%)
Number of pregnancies	N (%)
1	34 (39.5%)
2	34 (39.5%)
3	12 (14%)
4	6 (7%)
Baby’s gender	N (%)
Woman	36 (42%)
Man	50 (58%)
Lactation period	N (%)
<6 months	44 (51%)
6–12 months	22 (27%)
>6 months	19 (22%)

N—size; BMI—body mass index [kg/m^2^]; HBD—a week of pregnancy at the time of delivery; SE—standard deviation.

**Table 2 biology-14-00332-t002:** List of isolated bacterial strains from tested material.

Genera	Bacterial Strains Isolated from the Areola Before Breastfeeding	Bacterial Strains Isolated from Milk	Bacterial Strains Isolated from the Areola After Breastfeeding
*Streptococcus*	*S. parasanguinis*, *S. mitis_orale*, *S. pneumoniae*, *S. sanguinis*, *S. vestibularis*, *S. gordonii*, *S. salivarius*, *S. pseudopneumoniae*	*S. parasanguinis*, *S. mitis_orale*, *S. pneumoniae*, *S. sanguinis*, *S. vestibularis*, *S. gordonii*, *S. salivarius*, *S. pseudopneumoniae*, *S. infantis*, *S. gallolyticus*	*S. mitis_orale*, *S. vestibularis*, *S. parasanguinis*, *S. pseudopneumoniae*, *S. salivarius*, *S. vestibularis*, *S. gordonii*, *S. pneumoniae*
*Staphylococcus*	*S. hominis*, *S. aureus*, *S. epidermidis*, *S. haemolyticus*, *S. caprae*, *S. petrasii*, *S. capitis*, *S. warneri*	*S. hominis*, *S. aureus*, *S. epidermidis*, *S. haemolyticus* *S. caprae*, *S. petrasii*, *S. capitis*, *S. warneri*, *S. saprophyticus*, *S. borealis*, *S. lugdunensis*	*S. hominis*, *S. aureus*, *S. epidermidis*, *S. haemolyticus* *S. caprae*, *S. capitis*, *S. warneri*, *S. lugdunensis*
*Enterococcus*	*E. faecalis*	*E. faecalis*, *E. faecium*, *E. italicus*, *E. durans*	*E. faecalis*, *E. faecium*
*Micrococcus*	*M. luteus*	*M. luteus*	*M. luteus*
*Gemella*	*G. haemolysans*	*G. sanguinis*, *G. haemolysans*	*G. sanguinis*, *G. haemolysans*
*Dermacoccus*	*D. nishinomiyaensis*	*D. nishinomiyaensis*	
*Neisseria*	*N. flavescens_subflava group*, *N. sicca*, *N. cinerea*	*N. flavescens_subflava group*, *N. sicca*, *N. cinerea*	*N. flavescens_subflava group*, *N. sicca*, *N. cinerea*
*Veillonella*	*V. dispar*	*V. parvula*, *V. atypica*, *V. dispar*	*V. atypica*, *V. dispar*, *V. parvula*
*Corynebacterium*	*C. mucifaciens*, *C. durum*, *C. argentoratense*, *C. simulans*, *C. tuberculostearicum*	*C. tuberculostearicum*, *C. mucifaciens*, *C. kroppenstedtii*, *C. bovis*, *C. lipophiloflavum*, *C. accolens*, *C. durum*	*C. argentoratense*, *C. tuberculostearicum*, *C. pseudodiphtheriticum*, *C. bovis*, *C. kroppenstedtii*, *C. amycolatum*, *C. propinquum*
*Rothia*	*R. deutocariosa*, *R. mucilaginosa*, *R. aeria*, *R. terrae*, *R. amarae*, *R. kristinae*	*R. mucilaginosa*, *R. deutocariosa*, *R. amarae*, *R. kristinae*, *R. terrae*	*R. mucilaginosa*, *R. aeria*, *R. dentocariosa*
*Kocuria*	*K. rhizophila*	*K. rhizophila*	*K. rhizophila*
*Actinomyces*	*A. naeslundii*, *A. oris*, *A. graevenitzii*	*A. naeslundii*, *A. graevenitzii*	*A. naeslundii*
*Pseudomonas*	*P. oryzihabitans*, *P. stutzeri*, *P. luteola*, *P. massiliensis*, *P. fulva*, *P. monteilli*	*P. oleovorans*, *P. oryzihabitans*, *P. stutzeri*, *P. cedrina*, *P. fulva*, *P. vulneris*, *P. luteola*, *P. monteteilli*, *P. putida*, *P. koreensis*	*P. oryzihabitans*, *P. fulva*, *P. massiliensis*, *P. luteola*, *P. vulneris*, *P. stutzeri*
*Enterobacter*		*E. hormaechei*, *E. kobei* *E. bugandensis*, *E. ludwigii*, *E. roggenkampii*, *E. asburiae*	*E. kobei*
*Cutibacterium*	*C. avidum*, *C. flaccumfaciens*	*C. avidum*, *C. acnes*	*C. avidum*
*Moraxella*	*M. osloensis*	*M. osloensias*	*M. osloensis*, *M. catarrhalis*
*Klebsiella*	*K. pneumoniae*, *K. oxytoca*	*K. oxytoca*, *K. pneumoanie*	*K. oxytoca*
*Escherichia*	*E. coli*	*E. coli*	*E. coli*
*Acinetobacter*	*A. lwoffii*, *A. schindleri*, *A. johnsonii*, *A. junii*, *A. ursingii*	*A. ursingii*, *A. junii*, *A. baylyi*, *A. jonhsonii*, *A. lwoffii*, *A. parvus*, *A. pittii*, *A. baumannii*	*A. baylyi*, *A. lwoffii*, *A. lactucae*, *A. johnsonii*, *A. ursingii*, *A. junii*, *A. pittii*,
*Pantonea*		*P. agglomerans*, *P. septica*	*P. anthophila*
*Bacillus*	*B. cereus*	*B. cereus*	*B. cereus*
*Limosilactobacillus*		*L. reuteri*, *L. fermentum*, *L. brevis*	*L. brevis*
*Schaalia*	*S. odontolytica*	*S. odonolytica*	
*Leclercia*	*L. adecarboxylata*	*L. adecarboxylata*	
*Cytobacillus*	*C. firmus*	*C. firmus*, *C. horneckiae*	
*Lacticaseibacillus*	*L. rhamnosus*	*L. rhamnosus*, *L. paracasei*	
*Microbacterium*		*M. paraoxydans*	
*Psychrobacter*		*P. sanguinis*	
*Shewanella*		*S. oneidensis*	
*Stenotrophomonas*		*S. maltophila*	
*Raoultella*		*R. ornithinolytica*	
*Aeromonas*		*A. caviae*	
*Serratia*		*S. marcescens*	
*Buttiauxella*		*B. gaviniae*	
*Lactobacillus*		*L. gasseri*, *L. oris*, *L. crispatus*, *L. brevis*, *L. salivarius*, *L. rhamnosus*	
*Bifidobacterium*		*B. breve*	
*Empeadobacter*		*E. brevis*	

**Table 3 biology-14-00332-t003:** Comparison of the occurrence of bacteria in the studied groups using the chi-square test.

Genera	Areola Swab Before Breastfeeding/Human Milk *p*	Human Milk/Areola Swab After Breastfeeding *p*	Areola Swab After Breastfeeding/Areola Swab Before Breastfeeding *p*
*Streptococcus*	0.006	0.800	0.014
*Staphylococcus*	0.119	0.033	0.028
*Enterococcus*	0.001	0.001	0.999
*Micrococcus*	0.001	0.958	0.705
*Gemella*	0.200	0.095	0.090
*Dermacoccus*	0.848	0.900	0.990
*Neisseria*	0.034	0.100	0.012
*Veillonella*	0.001	0.637	0.783
*Corynebacterium*	0.969	0.043	0.020
*Rothia*	0.074	0.100	0.010
*Kocuria*	0.184	0.730	0.100
*Actinomyces*	0.400	>0.999	>0.999
*Pseudomonas*	0.001	0.001	<0.001
*Enterobacter*	0.027	0.700	0.999
*Cutibacterium*	0.848	0.001	0.913
*Moraxella*	0.782	0.802	0.238
*Klebsiella*	0.001	0.120	0.001
*Escherichia*	0.802	0.380	0.100
*Acinetobacter*	0.004	0.001	0.302
*Pantonea*	0.999	0.800	0.999
*Bacillus*	0.002	0.003	0.999
*Limosilactobacillus*	0.999	0.670	0.999
*Schaalia*	0.999	0.999	0.999
*Leclercia*	0.999	0.999	0.999
*Cytobacillus*	0.999	0.999	0.999
*Lacticaseibacillus*	0.999	0.999	0.999
*Microbacterium*	0.999	0.999	-
*Psychrobacter*	0.999	0.999	-
*Shewanella*	0.999	0.999	-
*Stenotrophomonas*	0.999	0.999	-
*Raoultella*	0.999	0.999	-
*Aeromonas*	0.999	0.999	-
*Serratia*	0.999	0.999	-
*Buttiauxella*	0.999	0.999	-
*Lactobacillus*	0.040	0.040	-
*Bifidobacterium*	0.999	0.999	-
*Empeadobacter*	0.999	0.999	-

*p* < 0.050 was considered statistically significant.

## Data Availability

Data will be made available upon request. Data are stored at the Department of Pathobiochemistry and Clinical Chemistry, Collegium Medicum in Bydgoszcz (Poland). Person responsible for providing data: Agnieszka Chrustek (mail: a.chrustek@cm.umk.pl).

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
