# Peer review of "Human Milk Microbiome from Polish Women Giving Birth via Vaginal Delivery—Pilot Study"

_biology, 2025, doi:10.3390/biology14040332_

Round 1
Reviewer 1 Report
Comments and Suggestions for Authors
This pilot study investigates the human milk (HM) microbiome in Polish women with normal BMI who delivered vaginally, using culture-based methods and MALDI-TOF MS for species identification. The topic is relevant, as understanding the HM microbiome’s role in infant gut colonization remains a critical area of research. However, the manuscript requires significant revisions to address methodological limitations, clarify interpretations, and strengthen the discussion. Below are specific concerns and recommendations:
- The reliance on culture-dependent methods limits the detection of unculturable or fastidious organisms, which are known to dominate microbial communities. This approach likely underestimates microbial diversity and biases results toward aerobic/facultative anaerobic species (e.g., Streptococcus, Staphylococcus). The authors should acknowledge this limitation and discuss how it impacts comparisons with studies using sequencing (e.g., 16S rRNA amplicon or shotgun metagenomics). Include a paragraph in the Discussion comparing culture-based findings with prior sequencing-based HM microbiome studies.
- The study only mentions Candida Were other fungi (e.g., Malassezia, Saccharomyces) screened? If not, clarify this limitation.
- The cohort (n=86) is reasonable for a pilot study, but demographic details (e.g., maternal age, parity, gestational age, antibiotic use during pregnancy/delivery, infant sex) are missing. These factors influence the HM microbiome and should be reported.
- No statistical tests are described to compare microbial diversity/composition between HM and areola/nipple swabs. Claims of “species specificity” to HM (e.g., Microbacterium, Shewanella) require statistical validation (e.g., PERMANOVA, LEfSe).
- LAB (Lactobacillus, Bifidobacterium) are highlighted as key beneficial genera, but their low prevalence (e.g., Bifidobacterium in only 3 HM samples) undermines claims about their significance.
- The hypothetical pathways of HM microbiome seeding (Fig. 1) are oversimplified.
- The lack of visual data (e.g., PCoA plots) makes it difficult to interpret microbial community structure.
- The link between HM microbiome and infant gut colonization is overstated. While LAB and HMOs are known to shape infant microbiota, the study does not measure infant outcomes or gut communities.
- The manuscript contains grammatical errors and ambiguous phrasing (e.g., “may influences” in Abstract).
- Outdated references (e.g., Dominguez-Bello et al., 2010) are cited, while recent HM microbiome studies are omitted.
- While the study provides a list of identified bacteria, the analysis is largely descriptive. The discussion could be strengthened by moving beyond simply listing species and delving deeper into the functional implications of the identified microbiome composition. Besides, the discussion is quite lengthy and at times becomes repetitive in describing the findings. It could be more focused and concise. For instance, elaborating on the functional roles of specific genera, such as Corynebacterium (known to produce amino acids like glutamic acid, nucleotides, and vitamins), would significantly enrich the discussion. Please expand on these functional aspects and cite doi: 10.1111/1749-4877.12763.
Author Response
Thank you for your comments on our manuscript, biology-3519428.
We have carefully revised the article according to your substantive suggestions. According to these constructive comments provided by you, we have carefully revised the article as described below. We believe that they have significantly improved the quality and value of our manuscript. We hope that you will find it a high-quality scientific work compatible with the high standards of the Biology journal. The detailed revisions are as follows:
Reviewer 1
- The reliance on culture-dependent methods limits the detection of unculturable or fastidious organisms, which are known to dominate microbial communities. This approach likely underestimates microbial diversity and biases results toward aerobic/facultative anaerobic species (e.g., Streptococcus, Staphylococcus). The authors should acknowledge this limitation and discuss how it impacts comparisons with studies using sequencing (e.g., 16S rRNA amplicon or shotgun metagenomics). Include a paragraph in the Discussion comparing culture-based findings with prior sequencing-based HM microbiome studies.
It has been added (lines 330).
- The study only mentions Candida Were other fungi (e.g., Malassezia, Saccharomyces) screened? If not, clarify this limitation.
Thank you for your valuable comment. The study presented was a pilot study and was based on the use of culture-based methods. Sheep blood agar was used in the study to allow growth of Candida species. Selective media for fungal culture were not used, which prevented the growth of other genera. Appropriate changes were made to the manuscript.
- The cohort (n=86) is reasonable for a pilot study, but demographic details (e.g., maternal age, parity, gestational age, antibiotic use during pregnancy/delivery, infant sex) are missing. These factors influence the HM microbiome and should be reported.
It has been added (line 196), table 1 has been modified
- No statistical tests are described to compare microbial diversity/composition between HM and areola/nipple swabs. Claims of “species specificity” to HM (e.g., Microbacterium, Shewanella) require statistical validation (e.g., PERMANOVA, LEfSe).
It has been added (Table 3)
- LAB (Lactobacillus, Bifidobacterium) are highlighted as key beneficial genera, but their low prevalence (e.g., Bifidobacterium in only 3 HM samples) undermines claims about their significance.
Thank you for your comment, of course we agree with the above fact. As we mentioned our research has a limitation in the form of culture methods, nevertheless we isolated 6 types of Lactobacillus from 15 women. We think that this is a good result considering our culture methods.
- The hypothetical pathways of HM microbiome seeding (Fig. 1) are oversimplified.
Thank you very much for this comment. We have improved Figure 1, which we believe is now more readable. We have also added an explanation regarding the hypothetical sources of the human milk microbiome.
It has been modified (line 129)
- The lack of visual data (e.g., PCoA plots) makes it difficult to interpret microbial community structure.
We decided to present it visually in the form of a Venn diagram, heatmaps and a dendrogram. In the future, after expanding the research, we would like to expand the data visualization using e.g. PCoA.
Figure 8 has been changed
Table 3 has been added
- The link between HM microbiome and infant gut colonization is overstated. While LAB and HMOs are known to shape infant microbiota, the study does not measure infant outcomes or gut communities.
Thank you very much for your valuable comment. Changes have been made – removed ,,These bacteria, together with human milk oligosaccharides (HMOs), support the growth and colonization of the child's gastrointestinal microbiota.”
- The manuscript contains grammatical errors and ambiguous phrasing (e.g., “may influences” in Abstract).
It has been changed
- Outdated references (e.g., Dominguez-Bello et al., 2010) are cited, while recent HM microbiome studies are omitted.
We are very grateful for your thorough assessment of the current literature. We have modified the article and added the latest scientific reports, thereby removing e.g., Dominguez-Bello et al., 2010.
Removed:
In contrast, babies born by cesarean section acquire bacteria that resemble the mother's skin microbiome, such as Staphylococcus spp. [53, 68].
Added:
Shao et al. found that cesarean sections reduce maternal bacteria transmission and increase colonisation by hospital-associated species like Enterococcus, Enterobacter, and Klebsiella. This indicates that delivery mode significantly impacts infant gut microbiota composition from neonatal period to infancy [Shao et al. 2019].
Shao Y, Forster SC, Tsaliki E, Vervier K, Strang A, Simpson N, Kumar N, Stares MD, Rodger A, Brocklehurst P, Field N, Lawley TD. Stunted microbiota and opportunistic pathogen colonization in caesarean-section birth. Nature. 2019 Oct;574(7776):117-121. doi: 10.1038/s41586-019-1560-1. Epub 2019 Sep 18. PMID: 31534227; PMCID: PMC6894937.
- While the study provides a list of identified bacteria, the analysis is largely descriptive. The discussion could be strengthened by moving beyond simply listing species and delving deeper into the functional implications of the identified microbiome composition. Besides, the discussion is quite lengthy and at times becomes repetitive in describing the findings. It could be more focused and concise. For instance, elaborating on the functional roles of specific genera, such as Corynebacterium (known to produce amino acids like glutamic acid, nucleotides, and vitamins), would significantly enrich the discussion. Please expand on these functional aspects and cite doi: 10.1111/1749-4877.12763.
Thank you very much for your valuable attention. We have modified the discussion. In order to strengthen the attractiveness of the manuscript, we have also added several important scientific reports, including the proposed manuscript doi: 10.1111/1749-4877.12763.
Reviewer 2 Report
Comments and Suggestions for Authors
1.The introduction provides a comprehensive background on the human milk microbiome but lacks a clearly stated hypothesis. The aim is mentioned as assessing the milk microbiome in Polish women, but the specific research questions or hypotheses (e.g., comparisons with other populations, factors influencing diversity) need explicit articulation to guide the reader.
2.The study design and sample collection process (e.g., milk expression timing, storage conditions) are described, but critical details are missing. For example:
Were the breast pumps sterilized before each use?
How were potential contaminants controlled during sample handling?
The rationale for selecting specific culture media (e.g., MRS agar for lactobacilli) is not justified.
3.While statistical tests (Pearson correlations, cluster analysis) are mentioned, the presentation of results lacks clarity. For instance:
The basis for selecting specific bacterial genera for correlation analysis is unclear.
The interpretation of heatmaps and dendrograms (Figures 7–8) is superficial; a more detailed explanation of clusters and their biological relevance is needed.
4.The isolation of Bacillus cereus (6% in human milk) raises concerns about contamination, as this pathogen is uncommon in human milk. The authors should address potential sources (e.g., environmental contamination during collection/storage) and discuss its clinical implications.
5.The study emphasizes Polish women, but the cohort is homogeneous (89.5% with higher education, 60% urban residents). This limits generalizability. A comparison with existing data from other regions (e.g., cited studies from Africa, Asia) would strengthen the discussion.
6.Relying solely on culture-dependent methods risks underestimating microbial diversity. The authors acknowledge this limitation but should elaborate on how unculturable or anaerobic species (e.g., Bifidobacterium) might have been underrepresented, affecting conclusions about the “core microbiome.”
7.Terms like “incendence” (likely “incidence”) and “L. crispantus” (possibly L. crispatus) contain typographical errors. The manuscript requires thorough language editing for clarity and consistency.
8.The discussion focuses heavily on confirming prior findings (e.g., Staphylococcus and Streptococcus dominance) but underaddresses novel results (e.g., L. crispatus isolation). The clinical significance of unique species (e.g., S. haemolyticus) and their potential transmission pathways (e.g., hospital environment) need deeper exploration.
Comments on the Quality of English Language
The English could be improved to more clearly express the research.
Author Response
Thank you for your comments on our manuscript, biology-3519428.
We have carefully revised the article according to your substantive suggestions. According to these constructive comments provided by you, we have carefully revised the article as described below. We believe that they have significantly improved the quality and value of our manuscript. We hope that you will find it a high-quality scientific work compatible with the high standards of the Biology journal. The detailed revisions are as follows:
Reviewer 2
1.The introduction provides a comprehensive background on the human milk microbiome but lacks a clearly stated hypothesis. The aim is mentioned as assessing the milk microbiome in Polish women, but the specific research questions or hypotheses (e.g., comparisons with other populations, factors influencing diversity) need explicit articulation to guide the reader.
It has been added (line 129)
2.The study design and sample collection process (e.g., milk expression timing, storage conditions) are described, but critical details are missing. For example:
Were the breast pumps sterilized before each use?
Yes, the breast pump was sterilized before each use. Before sterilization, each participant in the study disassembled all the elements that could come into contact with food or her skin, such as the funnel, bottle, valves and membranes. and washed them in water with added detergent, then thoroughly rinsed all the elements of the breast pump. In the research project, each of the participants in the study received a special bag from us for sterilizing the elements of the breast pump. According to the instructions on the sterilization bag - they poured 60 ml into the bag, then placed the elements of the breast pump in the bag, which was then carefully closed. Then they placed the bag in the microwave in an upright position and set the appropriate time and microwave power according to the following options:
- 1.5 minutes for 1100W power
- 3 minutes for 800W-1000W power
- 5 minutes for 500W-750W power
How were potential contaminants controlled during sample handling?
Samples were collected in sterile containers/swabs. All work related to bacterial culture and identification was performed in sterile conditions (under a laminar flow hood, using sterile loops, media).
The rationale for selecting specific culture media (e.g., MRS agar for lactobacilli) is not justified.
We decided to choose the described media due to our experience in identifying bacteria from milk. Columbia Agar with 5% addition of Sheep Blood is standardly used in clinical diagnostics and in quality control of food products. Due to the difficulties in isolating lactic acid bacteria, we also chose MRS agar for lactobacillus.
3.While statistical tests (Pearson correlations, cluster analysis) are mentioned, the presentation of results lacks clarity. For instance:
The basis for selecting specific bacterial genera for correlation analysis is unclear.
We performed correlations in the studied groups taking into account the bacteria that we isolated in a specific group. Therefore, for example, in milk there are more types of bacteria used, therefore there are more correlations. Significant correlations are listed in the text. In the smear before and after there are fewer of them. If there were no correlations between types of bacteria then they were not included.
Figure 8 has been changed
Table 3 has been added
- The interpretation of heatmaps and dendrograms (Figures 7–8) is superficial; a more detailed explanation of clusters and their biological relevance is needed.
It has been added
Figure 8 has been changed
Table 3 has been added
4.The isolation of Bacillus cereus (6% in human milk) raises concerns about contamination, as this pathogen is uncommon in human milk. The authors should address potential sources (e.g., environmental contamination during collection/storage) and discuss its clinical implications.
It has been modified (line 439)
5.The study emphasizes Polish women, but the cohort is homogeneous (89.5% with higher education, 60% urban residents). This limits generalizability. A comparison with existing data from other regions (e.g., cited studies from Africa, Asia) would strengthen the discussion.
Thank you for your attention. For us authors, the comparison of the human milk microbiome depending on the geographical location of the breastfeeding woman was also important to us, which is why we included this thread from the line …
6.Relying solely on culture-dependent methods risks underestimating microbial diversity. The authors acknowledge this limitation but should elaborate on how unculturable or anaerobic species (e.g., Bifidobacterium) might have been underrepresented, affecting conclusions about the “core microbiome.”
It has been modified (line 585)
7.Terms like “incendence” (likely “incidence”) and “L. crispantus” (possibly L. crispatus) contain typographical errors. The manuscript requires thorough language editing for clarity and consistency.
It has been changed
8.The discussion focuses heavily on confirming prior findings (e.g., Staphylococcus and Streptococcus dominance) but underaddresses novel results (e.g., L. crispatus isolation). The clinical significance of unique species (e.g., S. haemolyticus) and their potential transmission pathways (e.g., hospital environment) need deeper exploration.
Thank you very much for your attention. In this manuscript, we paid the most attention to the overall assessment of the human milk mycobiome, but also to the microbiological assessment of the areola and nipple skin before and after feeding. Despite the extensiveness of the discussion, we also included clinical information on L. crispatus and S. haemolyticus (line 360)
Reviewer 3 Report
Comments and Suggestions for Authors
Review of Human milk microbiome from Polsih women giving birth via vaginal delivery – pilot study, Chrustek et al.
This article provides an outstanding assessment of the microbiota associated with human milk and skin areas of the breast near feeding times from women who delivered their children vaginally. The composition of the milk microbiome has become an intense aspect of research given the connection of microbiome composition and neonatal development, especially neurological development in newborns. The microbiome is complex and differences not only in the types of bacteria but also their amounts can affect the health and development of children. The bacteria and human milk oligosaccharides provide a direct link to microbiome development in newborns. This article should be published after considering the following comments.
Major comments
- For the characterization of the cohort of women, the following variable need to be characterized and logged:
- Diet of women (vegetarian versus omnivore)
- Medications taken (especially any antibiotics recently and during, timframe)
- Medical history (current diseases and previous surgeries)
- Number of children (you have first and multi, but more definition should be done)
- Figure 6 should be accompanied with a supplemental table or figure outlining which bacteria fell into he different lobes of the Venn diagram. Also the total % in that figure is 99.5% and not 100%.
- Regarding any correlative studies and information, did any children have any medical problems? Were there any children that did not consume human milk early but had to transition from formula or from human milk to formula and did this correlate with any compositional analyses of the milk samples?
- Regarding sample composition, % of each sample for bacterial composition is given. Can numbers of bacteria be calculated so that numbers of each can be related to overall numbers of bacteria so that if an imbalance is identified it would have more resolution than on a % total?
- Figure 8 – the axes should be labeled and the labels on the figure are small and difficult to read.
- Minor grammar edits are needed throughout the manuscript.
Minor changes and edits needed throughout the manuscript
Author Response
Thank you for your comments on our manuscript, biology-3519428.
We have carefully revised the article according to your substantive suggestions. According to these constructive comments provided by you, we have carefully revised the article as described below. We believe that they have significantly improved the quality and value of our manuscript. We hope that you will find it a high-quality scientific work compatible with the high standards of the Biology journal. The detailed revisions are as follows:
Major comments
- For the characterization of the cohort of women, the following variable need to be characterized and logged:
- Diet of women (vegetarian versus omnivore)
- Medications taken (especially any antibiotics recently and during, timframe)
- Medical history (current diseases and previous surgeries)
- Number of children (you have first and multi, but more definition should be done)
It has been added (line 196), table 1 has been modified
- Figure 6 should be accompanied with a supplemental table or figure outlining which bacteria fell into he different lobes of the Venn diagram. Also the total % in that figure is 99.5% and not 100%.
We did not insert a table because we wanted to avoid repetition. In the work isolated bacteria are described in table 2 and also in the description. Of course we can add another table at any time.
Total % in venn diagram is 100%
- Regarding any correlative studies and information, did any children have any medical problems? Were there any children that did not consume human milk early but had to transition from formula or from human milk to formula and did this correlate with any compositional analyses of the milk samples?
The children were healthy and did not consume formula milk
- Regarding sample composition, % of each sample for bacterial composition is given. Can numbers of bacteria be calculated so that numbers of each can be related to overall numbers of bacteria so that if an imbalance is identified it would have more resolution than on a % total?
Due to the specificity of our study, we believe that the results presented in % best describe the occurrence of bacteria in our test material. In the future, we would like to expand the research and present it in a quantitative way.
- Figure 8 – the axes should be labeled and the labels on the figure are small and difficult to read.
It has been modified
Figure 8 has been changed
- Minor grammar edits are needed throughout the manuscript.
It has been modified
Reviewer 4 Report
Comments and Suggestions for Authors In this manuscript the diversity of bacteria in milk is analyzed in relation to the skin microbiota in polish women who underwent vaginal delivery, before and after breastfeeding. Cell culture and MALDI-TOF MS were used to identify the bacteria species. 120 species were identified, most of them belonging to the Staphylococcus and Streptococcus genera. Mechanisms of bacterial transfer and the role of milk microbiome are discussed. The results are a valuable contribution to the subject. However, some observations in order to improve the manuscript are listed below.Introduction
The introduction provides a comprehensive review of the state of the art regarding the origin of human milk microbiota, which is the subject of this investigation. Critical features of milk microbiota are well explained, such as its role in the colonization of infant gut microbiota, its dynamic as an ecological system, and some factors influencing the species diversity of human milk microbiome.
However, contrasting hypotheses on the origin of human milk microbiota are not presented. I suggest preparing a brief paragraph showing the current hypothesis on this subject, since it is useful not only to the readers but to the interpretation of the results of this work.
Methodology
Most of the studies on bacterial species identification are limited to the available techniques. Here, the main limitation is the bias to species that can grow in artificial culture media: between 50% to 90% of milk bacterial species do not grow in artificial media.
In relation to the statistical analysis, the plots and the inferential tests are well selected. However, a special consideration must be considered when working with ecological communities, specially for correlation analysis. The phylogenetic relationships are very important to understand the level of correlation between species. Closely related species may exhibit a high level of correlation due to a recent common ancestor and similar biological features; on the contrary, closely related species may exhibit low level of correlation due to recent diversification on ecological niche. Similarly, phylogenetically distant species may show low correlation due to a distant common ancestor driving to high degree of phenotypic divergence, or high correlation due to convergence evolution in relation to the ecological niche.
To overcome this problem, the phylogenetic distance between pairs of species must be considered to adjust correlation models, for example using the Mantel test or phylogenetic independent methods such as PGLS; Phylogenetic Generalized Least Squares. Another alternative is to use phylogenetic co-occurrence analysis to assess whether species associations follow non-random evolutionary patterns. Databases such as GenBank can be used to obtain 16S rRNA sequences and reconstruct phylogenetic trees.
Results
The plots and statistical tests are well selected and presented. New results derived from the comparative analyses suggested in the commentaries on methodology should improve this section.
Discussion
The discussion offers an adequate interpretation and reflection on the results. However, I think that a more evolutionary perspective could improve this section, both in relation to the cluster of species found in this population and as well the findings in other mammal species. The next two references are suggested:
Muletz-Wolz, C. R., Kurata, N. P., Himschoot, E. A., Wenker, E. S., Quinn, E. A., Hinde, K., Power, M. L., & Fleischer, R. C. (2019). Diversity and temporal dynamics of primate milk microbiomes. American journal of primatology, 81(10-11), e22994. https://doi.org/10.1002/ajp.22994
Baniel, A., Petrullo, L., Mercer, A., Reitsema, L., Sams, S., Beehner, J. C., Bergman, T. J., Snyder-Mackler, N., & Lu, A. (2022). Maternal effects on early-life gut microbiota maturation in a wild nonhuman primate. Current biology : CB, 32(20), 4508–4520.e6. https://doi.org/10.1016/j.cub.2022.08.037
Author Response
Thank you for your comments on our manuscript, biology-3519428.
We have carefully revised the article according to your substantive suggestions. According to these constructive comments provided by you, we have carefully revised the article as described below. We believe that they have significantly improved the quality and value of our manuscript. We hope that you will find it a high-quality scientific work compatible with the high standards of the Biology journal. The detailed revisions are as follows:
I suggest preparing a brief paragraph showing the current hypothesis on this subject, since it is useful not only to the readers but to the interpretation of the results of this work.
It has been added (line 129)
Define more explicitly which aspects of the human milk microbiota remain unexplored and how this study contributes to filling those gaps.
It has been added (583)
Address the limitations of culture-based techniques and discuss potential biases introduced by this approach.
It has been added (line 585)
Include a comparative framework, such as phylogenetic analysis, to better understand bacterial associations in human milk.
Thank you very much for your attention, of course we fully agree that phylogenetic analysis would improve our manuscript. Unfortunately, due to the lack of molecular analyses we are not able to extend our results with a phylogenetic tree or use Databases such as GenBank. In the future we intend to extend our studies and then we will certainly use the suggested models.
In the work we included a dendrogram as a cluster analysis. Due to the breeding methods used and not molecular ones, we are not able to present a phylogenetic analysis. At present, these are our pilot studies, therefore in the future we would like to expand the research, which is associated with making the results more attractive in the form of phylogenetic analysis.
The discussion offers an adequate interpretation and reflection on the results. However, I think that a more evolutionary perspective could improve this section, both in relation to the cluster of species found in this population and as well the findings in other mammal species.
Thank you for your suggestions for literature. The discussion has been modified. In our work, due to limitations, phylogenetic analysis was not performed, so we did not use the suggested publications. However, in the future we will definitely include phylogenetic analysis and suggested publications.
Round 2
Reviewer 2 Report
Comments and Suggestions for Authors
The authors have made substantial revisions to the manuscript, addressing many of the initial concerns and improving the clarity of the methods, results, and figures. The inclusion of additional data, such as the Venn diagram and heatmaps, strengthens the presentation of microbial diversity across samples. However, the Discussion section remains disorganized and requires significant restructuring to enhance coherence, focus, and alignment with the study’s objectives. Below are specific critiques and suggestions for improvement:
1.The Discussion jumps between topics (e.g., probiotic potential of isolated bacteria, pathogenic concerns, geographical comparisons) without a logical flow. A clear framework (e.g., discussing results in the order of hypotheses, key findings, limitations, and implications) would improve readability.
Suggestion: Organize the Discussion into subsections (e.g., “Core Microbiota of Human Milk,” “Potential Pathogens and Contamination Risks,” “Geographical and Methodological Influences,” “Limitations and Future Directions”).
2.Excessive focus on bacterial species with minimal relevance to the study’s core findings (e.g., detailed descriptions of Acinetobacter baumannii or Bacillus cereus) distracts from the main themes.
Suggestion: Prioritize discussing results that directly address the study’s aims (e.g., transmission routes, shared vs. unique species across samples, implications for infant gut colonization).
3.Multiple paragraphs redundantly describe the probiotic roles of Lactobacillus and Bifidobacterium or rehash methodological limitations without adding new insights.
Suggestion: Consolidate repetitive content and focus on novel contributions (e.g., the isolation of L. crispatus or unique species in Polish women).
4.The three hypothetical sources of the milk microbiome (entero-mammary route, skin transfer, infant retrograde flow) are mentioned briefly but not systematically evaluated against the results.
Suggestion: Explicitly map the isolated species (e.g., Streptococcus from infant oral cavity, Staphylococcus from skin) to the proposed hypotheses and discuss which pathways are best supported by the data.
Author Response
Thank you for your suggestion. We have tried to improve the discussion as much as we could. We hope that the corrections will enrich the manuscript.
1.The Discussion jumps between topics (e.g., probiotic potential of isolated bacteria, pathogenic concerns, geographical comparisons) without a logical flow. A clear framework (e.g., discussing results in the order of hypotheses, key findings, limitations, and implications) would improve readability.
Suggestion: Organize the Discussion into subsections (e.g., “Core Microbiota of Human Milk,” “Potential Pathogens and Contamination Risks,” “Geographical and Methodological Influences,” “Limitations and Future Directions”).
It has been changed.
2.Excessive focus on bacterial species with minimal relevance to the study’s core findings (e.g., detailed descriptions of Acinetobacter baumannii or Bacillus cereus) distracts from the main themes.
Suggestion: Prioritize discussing results that directly address the study’s aims (e.g., transmission routes, shared vs. unique species across samples, implications for infant gut colonization).
Thanks for your suggestions. The discussion contains different bacteria due to suggestions from other reviewers.
3.Multiple paragraphs redundantly describe the probiotic roles of Lactobacillus and Bifidobacterium or rehash methodological limitations without adding new insights.
Suggestion: Consolidate repetitive content and focus on novel contributions (e.g., the isolation of L. crispatus or unique species in Polish women).
It has been modified
4.The three hypothetical sources of the milk microbiome (entero-mammary route, skin transfer, infant retrograde flow) are mentioned briefly but not systematically evaluated against the results.
Suggestion: Explicitly map the isolated species (e.g., Streptococcus from infant oral cavity, Staphylococcus from skin) to the proposed hypotheses and discuss which pathways are best supported by the data.
It has been added (line 378, 430)